# Influence of Systemic Therapy on the Expression and Activity of Selected STAT Proteins in Prostate Cancer Tissue

**DOI:** 10.3390/life12020240

**Published:** 2022-02-06

**Authors:** Celina Ebersbach, Alicia-Marie K. Beier, Pia Hönscheid, Christian Sperling, Korinna Jöhrens, Gustavo B. Baretton, Christian Thomas, Ulrich Sommer, Angelika Borkowetz, Holger H. H. Erb

**Affiliations:** 1Department of Urology, Technische Universität Dresden, 01307 Dresden, Germany; Celina.Ebersbach@uniklinikum-dresden.de (C.E.); AliciaMarie.Beier@uniklinikum-dresden.de (A.-M.K.B.); Christian.thomas@uniklinikum-dresden.de (C.T.); Angelika.Borkowetz@uniklinikum-dresden.de (A.B.); 2Mildred Scheel Early Career Center, Department of Urology, Medical Faculty and University Hospital Carl Gustav Carus, Technische Universität Dresden, 01307 Dresden, Germany; 3Institute of Pathology, Universitätsklinikum Carl Gustav Carus Dresden, 01307 Dresden, Germany; Pia.Hoenscheid@uniklinikum-dresden.de (P.H.); chrisitan.sperling@uniklinikum-dresden.de (C.S.); Korinna.Joehrens@ukdd.de (K.J.); Gustavo.Baretton@uniklinikum-dresden.de (G.B.B.); Ulrich.Sommer2@Uniklinikum-Dresden.de (U.S.); 4National Center for Tumor Diseases Partner Site Dresden and German Cancer Center Heidelberg, 69120 Heidelberg, Germany; 5Tumor and Normal Tissue Bank of the University Cancer Center (UCC), University Hospital and Faculty of Medicine, Technische Universität Dresden, 01307 Dresden, Germany

**Keywords:** STAT3, STAT5, STAT6, androgen deprivation therapy, novel hormonal therapy, chemotherapy, HSPC, CRPC, therapy resistance

## Abstract

Signal Transducer and Activator of Transcription (STAT) proteins have been identified as drivers of prostate cancer (PCa) progression and development of aggressive castration-resistant phenotypes. In particular, STAT3, 5, and 6 have been linked to resistance to androgen receptor inhibition and metastasis in in vitro and in vivo models. This descriptive study aimed to validate these preclinical data in tissue obtained from patients with PCa before and while under androgen-deprivation therapy. Therefore, STAT3, 5, and 6 expressions and activity were assessed by immunohistochemistry. The data revealed that STAT3 and 5 changed in PCa. However, there was no relationship between expression and survival. Moreover, due to the heterogeneous nature of PCa, the preclinical results could not be transferred congruently to the patient’s material. A pilot study with a longitudinal patient cohort could also show this heterogeneous influence of systemic therapy on STAT3, 5, and 6 expressions and activity. Even if the main mechanisms were validated, these data demonstrate the urge for better patient-near preclinical models. Therefore, these data reflect the need for investigations of STAT proteins in a longitudinal patient cohort to identify factors responsible for the diverse influence of system therapy on STAT expression.

## 1. Introduction

Prostate cancer (PCa) is one of the most common cancers in men, with an estimated number of over 1.4 million new cases worldwide in 2020. In addition, with an estimated 375,000 new deaths worldwide, PCa is also one of the leading causes of death in men [1]. Organ-confined PCa (TMN stage 1 and 2) is treated with curative intent by radical prostatectomy or radiation therapy [2]. For these patients, 5-year survival is 100% [3]. However, in patients with metastasized PCa (TMN stage 3 and 4) 5-year survival is ~30% [3]. Since PCa is androgen dependent in its growth, first-line therapy includes androgen-deprivation therapy (ADT), which is indicated for metastasized hormone-sensitive PCa (mHSPC) and can be combined with taxan-based chemotherapy (CTx) or novel hormonal therapy (NHT) as apalutamide, abiraterone, or enzalutamide [2,4]. Unfortunately, resistance to ADT inevitably occurs, and after a medium response duration of 18 months, tumors progress to castration-resistant PCa (CRPC). Treatment options for non-metastasized CPRC are the antiandrogens apalutamide, darolutamide, and enzalutamide [4]. Treatment for metastasized CRPC (mCRPC) includes taxane-based chemotherapy with docetaxel or cabazitaxel, abiraterone, and the antiandrogen enzalutamide [4]. The PARP-inhibitor Olaparib was recently approved for patients with mCRPC and alteration of one or more homologous recombination repair genes [4]. Unfortunately, despite recent improvements in treatment options, patients suffering from mCRPC only have a median life expectancy of 19 months due to tumor progress and emerging therapy resistance [5].

The Signal Transducers and Activators of Transcription (STATs) are a group of transcription factors, which reportedly are involved in those life-threatening processes in PCa [5]. The group contains seven members, of which STAT3, STAT5, and STAT6 are involved in tumor progress, metastasis, and therapy resistance.

STAT3 is highly expressed and active in advanced PCa and PCa bone metastases [6,7]. In addition, the transcription factor is involved in the transdifferentiation processes of the PCa cell line LNCaP to ADT and antiandrogen-resistant neuroendocrine PCa cells [8,9]. Moreover, in vitro studies revealed that crosstalk between the STAT3 and the androgen receptor (AR) axis potentiates the androgen-dependent transactivation of the AR and therefore mediates resistance to the antiandrogen enzalutamide, which can be reversed by STAT3 inhibition [10,11]. 

STAT5 expression correlates with high Gleason scores (GS) and predicts early recurrence of PCa after radical prostatectomy [12,13]. Furthermore, Thomas et al. has reported that ADT leads to increased STAT5 expression in PCa tissue [12]. Moreover, STAT5 influences the AR activity directly by regulating the AR stability, an important factor in antiandrogen response [12,14,15]. In addition, increased STAT5 expression and transactivity have been revealed in enzalutamide-resistant cell lines and siRNA-mediated STAT5-knockdown resensitized the enzalutamide resistant MR49F cells to enzalutamide [16,17]. 

Both STAT3 and STAT5 have been reported to be increased in docetaxel-resistant cell lines compared to their sensitive controls. However, their function in docetaxel-resistant cells has not yet been investigated [18].

STAT6 expression is elevated in PCa compared to benign prostate, while STAT6 expression correlates with higher GS and larger tumor size [19,20]. In vitro experiments revealed a possible role of STAT6 in metastasis. However, involvement in therapy resistance has not yet been described [19].

As most findings of STAT proteins in therapy resistance have been revealed in cell line experiments, data about expression and activity in therapy-resistant PCa tissue are limited. Therefore, this descriptive study aimed to examine the expression and activity of STAT3, STAT5, and STAT6 in PCa patients undergoing systemic therapies, including ADT, enzalutamide, abiraterone, and docetaxel.

## 2. Materials and Methods

### 2.1. Patient Material and Immunohistochemistry (IHC)

Patients’ samples were selected from the Tumor and Normal Tissue Bank of the University Cancer Center Dresden. The Ethics Committee of the Technische Universität Dresden approved the use of archived material (Study no. EK59032007). According to statutory provisions, written consent was obtained from all patients and documented in the database. The cohort contained 154 formalin-fixed and paraffin-embedded (FFPE) tissue specimens of 97 PCa patients and 32 patients with benign prostate hyperplasia (BPH) undergoing TURP. Tissue blocks were cut in serial sections of 1–2 µm thickness; sections were deparaffinized with BenchMark XT (Ventana Medical Systems, Oro Valley, AZ, USA) and then exposed to a heat-induced epitope retrieval. The antibodies listed in Table 1 were used for staining, followed by counterstaining with hematoxylin, dehydration, and mounting of the slides. IHC was evaluated by the department of pathology Dresden using the Remmele Score [21]. Therefore, the immunoreactivity score (IRS) was calculated according to the following parameters: staining intensity was scored 0–3 (0 = absent, 1 = low intensity, 2 = average intensity, 3 = intense). The percentage of positively stained cells was scored 0–4 (0 = absent; 1 ≤ 10%; 2 ≤ 50%; 3 ≤ 80%; 4 > 80%). Both scores were multiplied to obtain the IRS, ranging from 0–12. The percentage of cells with nuclear STAT was assessed to investigate STAT activity. Therefore, the activity of unphosphorylated and phosphorylated STAT was included in the study. IHC slides were digitalized using a Pannoramic Scan II (3DHistech LTD, Budapest, Hungary) scanner. IRS data from the different cohorts were displayed as box and whisker diagrams (min to max). The IRS data from the longitudinal patient cohort were displayed as individual data points. The corresponding IRS values of one patient have been connected with a line. 

### 2.2. Statistical Analysis

Prism 9.3.1 (GraphPad Software, San Diego, CA, USA) was used for all statistical analyses. Differences between treatment groups were analyzed using ordinary one-way ANOVA or Student’s *t*-test. *p*-values of ≤0.05 were considered statistically significant. A Kaplan–Meier estimate was used for overall survival analysis. The Pearson correlation coefficient (r) was calculated and interpreted by the guidelines suggested by Schober et al. for correlation analysis [21]. All differences highlighted by asterisks were statistically significant as encoded in figure legends (* *p* ≤ 0.05; ** *p* ≤ 0.01; *** *p* ≤ 0.001).

## 3. Results

### 3.1. Description of the Patient’s Cohort

For this study, 97 PCa patients (26 HSPC patients, 71 ADT patients, of which 25 received ADT only, 33 received ADT + NHT, and 13 received ADT + CTx) and 32 BPH patients (a non-malignant control cohort) were retrospectively selected. In the used cohort, ADT was induced by treatment with buserelin, triptorelin, degarelix, or leuprorelin. For ADT + NHT, ADT combined with enzalutamide or abiraterone was administered. For ADT + CTx, ADT combined with docetaxel or cabazitaxel was administered. The treatment regimen was used for at least one month. Patients were recruited between 2011 and 2020. In total, 154 FFPE PCa tissue specimens were taken after palliative TURP. Baseline patient characteristics are summarized in Table 2.

Kaplan–Meier analysis revealed a median overall survival (OS) of 201 months (Hazard Ratio, 95% Confidence Interval: 1.5; 0.8–2.9) for patients with HSPC and 136 months (0.7; 0.3–1.3) for patients undergoing ADT (Figure 1A). Patients receiving additional treatment (Figure 1B) with NHT (OS: 187 months) or CTx (OS: 187 months) had a non-significant increase in OS compared to the ADT patients (OS: 128 months). 

Therefore, treatment did not influence OS in this cohort. 

### 3.2. Change in STAT 3, 5, and 6 Expressions in PCa Compared to BPH 

STAT protein expression was reported to change during malignant transformation of the prostate [5]. Therefore, to evaluate STAT3 (Figure 2), 5 (Figure 3) and 6 protein expression (Figure 4) patterns in BPH and PCa, tissue of the presented patient cohort was investigated using the Remmele Score. Therefore, an immune reactivity score (IRS) was calculated using the grade of staining intensity and the fraction of positive cells was evaluated (Figure 5) [21].

The IRS evaluation of STAT3 (Figure 5A) revealed a significant downregulation of STAT3 in PCa (mean IRS ± SD: 4.9 ± 3.0) compared to BPH (mean IRS ± SD: 7.1 ± 2.5). In contrast, IRS evaluation of STAT5 showed a significant increase in STAT5 in PCa (mean IRS ± SD: 5.2 ± 2.1) compared to BPH (mean IRS ± SD: 3.9 ± 2.1). Assessment of STAT6 showed no significant difference in the STAT6-IRS between the PCa (mean IRS ± SD: 2.0 ± 1.4) and BPH (mean IRS ± SD: 2.9 ± 2.2) groups. Correlation analysis (Figure 5B) showed no significant correlation between the STAT protein IRS scores in PCa. To assess the activity of the investigated STATs, the percentage of nuclear localization was examined in the malignant areas. The analysis of nuclear STAT revealed a significant decrease in nuclear localization in STAT3 and STAT5, whereas STAT6 showed almost no nuclear localization at all (Figure 5C). Correlation analysis revealed no correlation between the investigated STAT proteins IRS scores and activity in PCa (Appendix A). Kaplan–Meier survival analysis showed that STAT3, 5, or 6 expressions did not affect OS for all examined STAT proteins. STAT3 activity did not affect OS (Figure 5G). Even if not significant, patients with low STAT5 activity (Figure 5H) had a longer OS than patients with high STAT5 activity (143 months vs. 77 months). No STAT6 activity could be detected, so no Kaplan–Meier estimation between high and low activity could be performed (Figure 5I).

Taken together, STAT 3 and 5 expression and activity changes in PCa, but does not significantly influence OS. 

### 3.3. Influence of Systemic Therapy on STAT3, 5, and 6 Protein Levels in PCa

Systemic therapy such as ADT, NHT, or CTx was reported to change STAT expression and activity and promote tumor progression and therapy resistance [5,10,12,16,17,22]. Therefore, the ADT PCa cohort was divided into HSPC and ADT subgroups to verify these findings. As a result of this subgrouping, the ADT subgroup includes patients receiving ADT, ADT+ NHT, and ADT + CTx. Even if not significant, IRS evaluation revealed a change from HSPC to the ADT subgroup from 4.6 ± 2.5 to 5.0 ± 3.1 for STAT3, 4.5 ± 3.0 to 5.4 ± 2.7 for STAT5, and 3.0 ± 2.4 to 2.9 ± 2.2 for STAT6 (Figure 6A). In addition, no significant but minor increase in the nuclear localization of STAT3 and 5 could be detected, whereas no nuclear localization could be seen for STAT6 (Figure 6B). However, expression and activity are highly heterogeneous in the evaluated cohorts (Figure 6). 

To investigate if further treatment with NHT or CTx affects STAT3, 5, or 6 expressions, the ADT subgroup was further divided into ADT, ADT + NHT, and ADT + CTx (Figure 7). IRS evaluation of STAT3 did not show any change after treatment with ADT + NHT and ADT + CTx (Figure 7C). Additional therapy with NHT or CTx did not change STAT3 nuclear localization compared to the ADT-only subgroup (Figure 7B). Additionally, the assessment of STAT5 (Figure 7C,D) and STAT6 (Figure 7C,D) did not reveal any change in expression or activity between the three subgroups. Analysis of the public PRAD SU2C 2019 cohort also indicated no change in STAT3, 5, and 6 expressions in the ADT, ADT + NHT, and ADT + CTx subgroups (Appendix A). 

In summary, no association between system therapy and STAT expression and activity can be established.

### 3.4. Influence of Systemic Therapy on STAT3, 5, and 6 Protein Levels in PCa

As a general comparison of the cohorts did not show any difference in STAT3, 5, and 6 expressions between the ADT subgroups, a pilot study was performed with TURP tissue of six patients obtained before and under ADT. These tissue specimens were investigated for STAT levels and activity. STAT3 expression was elevated after ADT treatment in three out of six patients (Figure 8A), whereas two patients decreased STAT3 levels. Likewise, STAT5 expression was increased in two out of six cases, whereas in one patient, STAT5 was reduced (Figure 8B). In contrast, STAT-6 expression was elevated and decreased in two patients (Figure 8C).

For STAT3, investigation of the STAT activity revealed a decrease in nuclear localization in three out of six patients and only one increase in nuclear localization after ADT (Figure 9B). STAT5 nuclear localization decreased in two patients and increased in only one patient (Figure 9B). STAT6 showed no nuclear localization, independent of treatment status (Figure 9C).

## 4. Discussion

STAT proteins influence multiple biological processes such as immune response, mitogenesis, wound healing, cell survival, and cell growth [5,23]. After activation, the transcription factors transmit their signals from the cell membrane through growth factors, cytokines, or hormones into the nucleus. They bind to specific response elements on the DNA in the nucleus, thereby inducing gene transcription [23]. Therefore, the STAT proteins interfere with several health conditions such as autoimmune diseases and cancer, including PCa. In PCa, the STAT protein family promotes tumor progression and mediates therapy resistance by regulating oncogene and pro-survival gene transcription (e.g., BCL2L1, MCL1, MYC, CCND1) [5]. However, most findings of STAT proteins in PCa are obtained from preclinical models, and data about expression and activity in advanced and therapy-resistant PCa tissue are limited. This study investigated three specific STAT proteins, STAT3, STAT5, and STAT6, in PCa tissue from patients undergoing systemic therapies. The effects of systemic treatment on the selected STAT proteins shown in the preclinical models are evident and well described. These changes should already be reflected in a small cohort as presented in this study [5].

STAT3 is the most investigated STAT protein in PCa. It is aberrantly activated in ~50% of PCa patients and modulates androgen receptor expression and activity [5,6,7,8]. STAT3 is activated by several proinflammatory cytokines, including interleukin-6 (IL-6) which is elevated in PCa patients and is the driver of PCa progression [24,25,26]. In vitro data support the oncogenic and growth-promoting role of IL-6 and STAT3 [11,27]. Furthermore, increased activated STAT-3 is associated with clinicopathological parameters such as high T-stage and increased Gleason score [28,29]. Moreover, IHC analysis of CRPC revealed increased STAT3 expression compared to BPH tissue [30]. In contrast to these findings, the IHC analysis revealed a significant decrease in STAT3 expression and activity in primary PCa tissue compared to BPH. This finding is in line with the studies published by Pencik and colleagues [31]. The group demonstrated that STAT3 exerts a tumor-suppressive function by activating senescence via the p19^ARF^–Mdm2–p53 axis at an early stage of PCa development. Based on these data, STAT3 may also play a tumor suppressor role and promote different functions depending on the tumor stage described for STAT1 [32]. These data are supported by Handle et al., revealing an increase in STAT3 activity but not STAT3 expression in castration-resistant and androgen-responsive patient-derived xenografts [10].

STAT5, which refers to the proteins STAT5a and STAT5b, was reported to play an essential role in the progression of PCa to CRPC and the development of enzalutamide resistance [5]. Similar to STAT3, there is evidence for an increase in STAT5 in CRPC [12,30]. Here, an elevated STAT5 level could be confirmed in primary PCa tissue compared to BPH. Furthermore, the growth hormone–STAT5 axis was linked to malignant transformation in several cancers, including PCa [33,34,35,36,37]. Therefore, the increased STAT5 indicates the involvement of the transcription factor in the oncogenic transformation of prostate cells. However, more investigations are necessary to validate this hypothesis. 

STAT6 in vitro data suggested a role of the IL-4/STAT6 in disease relapse by providing a favorable niche for the clonogenic growth of tumor-inducing PCa cells [19]. In PCa, STAT6 has increased expression and activity in malignant regions compared to adjective normal areas [19]. However, compared to BPH tissue, the PCa assessed here did not show a difference in STAT6 levels. Moreover, no activation could be detected in both cohorts. This result agrees with the role of the IL-4/STAT6 axis in metastasis as described earlier [19,38,39]. Moreover, activation of STAT6 was linked to high serum levels of IL-4 or increased M2 macrophage infiltration, which has not been assessed in the present study [40,41]. 

Kaplan–Meier estimations revealed a role of STAT proteins in biochemical recurrence and OS of PCa patients [5]. High levels of activated STAT3 and high STAT3 expression are linked to biochemical recurrence and shorter OS [42,43]. Increased STAT5 is associated with lower disease-free survival after RP in PCa [44]. Only high STAT5 activity was linked to a shorter OS in the present study. However, it must be noted that possible statistical effects are hidden due to the small cohort size. Additionally, treatment failure and biochemical recurrence were chosen as endpoints in most cases. Several studies used the phosphorylated STAT protein and not the localization. However, when evaluating phosphorylated STAT3 and STAT5, the activity of unphosphorylated STATs is not taken into account [45].

Multiple reports have associated STAT proteins with the development of therapy resistance. Preclinical studies have linked STAT3 to ADT, enzalutamide, and docetaxel resistance in PCa [5]. It is suggested that under ADT, active STAT3 enhances AR activity in the presence of low levels of androgens and therefore enhances PCa progression [10,46,47]. In this study, increased STAT3 levels and activity could also be observed in patients receiving ADT. This result confirms the role of STAT3 in PCa progress [11,27,28,29]. In vitro studies also revealed a function of the STAT3 signal pathway in enzalutamide and docetaxel resistance [11,48,49]. Here we could not show a further increase in STAT3 expression and activity in PCa tissue obtained from patients after receiving NHT or CTx. However, the absence of further change in STAT3 expression and activity does not rule out involvement of STAT3 in NHT or CTx resistance. 

STAT5 promotes resistance to ADT and the development of the aggressive CRPC [12,50]. Therefore, it is suggested that STAT5 stabilizes the AR and thus promotes AR activity and PCa progression. AR stability was identified as a main regulator of AR after androgen deprivation and antiandrogen treatment [14]. In line with the data from Thomas and colleagues, an increase in STAT5 levels and activity could be observed in patients after receiving ADT [12]. The amplification of the STAT5 gene may partially explain this increase during ADT [51]. Enzalutamide has also been described to increase STAT5 activity by activating an AR-induced JAK2/STAT5 feed-forward loop [16,52]. Additionally, increased STAT5 levels have been demonstrated in docetaxel-resistant Du145 cells compared to their docetaxel-sensitive controls [18]. However, additional treatment with NHT or CTx increased neither STAT5 levels nor activity. 

STAT6 has not been linked to the development of therapy resistance in PCa so far. Additionally, treatment with ADT or ADT combined with NHT and CTx did not influence STAT6 activity and expression in this study.

Tumor multifocality and heterogeneity are among the biggest challenges in primary PCa research and management. Therefore, many findings from preclinical models cannot be directly applied to primary tumor material, and statistical significance can hardly be achieved. Additionally, the data presented in this study reveal high heterogeneous expression and activity of the investigated STAT proteins. This heterogeneity may mask treatment-induced changes previously seen in in vitro and in vivo experiments [5]. For this reason, PCa tissue from six patients obtained from palliative TURP before and during ADT was analyzed for STAT3, 5, and 6 expression and activity changes. None of the investigated STATs showed a homogenous response to ADT in this small cohort. This finding is in line with the data from Handle et al. or Bishop et al. revealing multiple independent mechanisms in developing enzalutamide-resistant LNCaP cells and demonstrating numerous response possibilities to one treatment despite the same cellular background [53,54]. 

Preclinical cell models have been used to investigate signal pathways and mimic tumor behavior to identify new therapeutic strategies. In particular, cell line models such as LNCaP cells have been used as they are easy to handle and the results are highly reproducible [55,56]. However, most cell line models have been intensively cultured over decades, and they have changed due to accumulations of mutations and chromosomal aberrations [56,57]. In addition, culturing them in 2D without the supporting tumor microenvironment led to adaption to the cell culture conditions not representing their natural environment [55]. Due to these adaptions, most findings identified in 2D cell culture models cannot be translated directly into natural tumor biology. Therefore, more complex and patient near cell models are mandatory to increase the physiological relevance of data identified in basic research. Among other model systems, ex vivo tissue slice models, organoid models, and patient-derived xenografts are highly recommended to maintain physiological relevance. Although these models have shown to be expensive and difficult to handle, they represent the tumor heterogeneity and physiological relevance best. For example, it could be demonstrated that the tumor environment alters tumor metabolism as well as glutamine and glucose dependence. Therefore, findings identified in cell line models need to be validated in a patient near complex model to be described as a novel mechanism. 

One limitation of this study is the low sample number, especially from the longitudinal patient cohort. Moreover, cancer-specific survival and mortality would have been desirable. Finally, expression data about the known STAT targets gene would also have been an excellent addition to estimating the tissue’s STAT activity. 

## 5. Conclusions

Most STAT proteins’ functional role in PCa progression was obtained in preclinical in vitro and in vivo models. Therefore, this study attempted to transfer the preclinical observations to patient material. Although there was no relationship between expression and survival, the data revealed that STAT 3 and 5 changed in PCa. The biggest hurdle to transfer the in vitro and in vivo data to the patient situation is the tumor heterogeneity and the different tumor response to the treatments, which can only be represented to a limited extent in preclinical models. Therefore, these data show the need for investigations of STAT3 and 5 in a longitudinal patient cohort to identify factors responsible for the diverse influence of system therapy on STAT3, 5, and 6 expressions.

## Figures and Tables

**Figure 1 life-12-00240-f001:**
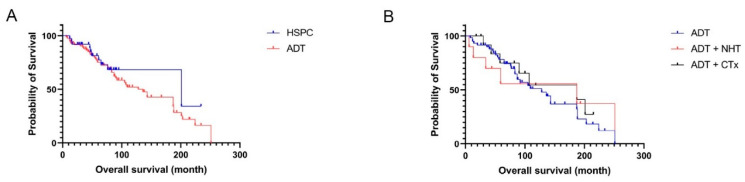
Kaplan–Meier analysis of the PCa cohort. (**A**) Kaplan–Meier analysis of the HSPC vs. the ADT therapy cohort; (**B**) Kaplan–Meier analysis of the different ADT cohorts.

**Figure 2 life-12-00240-f002:**
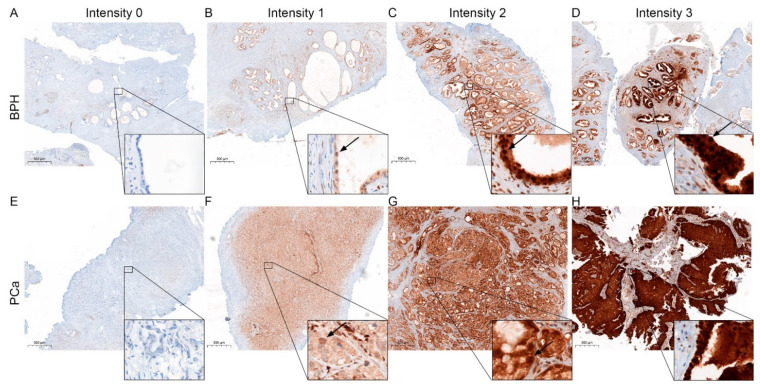
Representative STAT3 staining in BPH and PCa. (**A**–**D**) Immunohistochemical staining for STAT3 of representative benign tissue with different staining intensities cores. Scale bar = 500 μM. (**E**–**H**) Immunohistochemical staining for STAT3 of representative malignant tissue with different staining intensities cores. Arrows mark representative nuclear localization. Scale bar = 500 μM.

**Figure 3 life-12-00240-f003:**
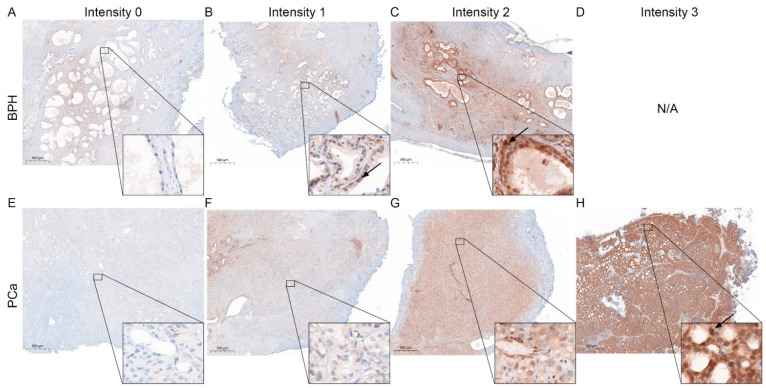
Representative STAT5 staining in BPH and PCa. (**A**–**D**) Immunohistochemical staining for STAT5 of representative benign tissue with different staining intensities cores. Scale bar = 500 μM. (**E**–**H**) Immunohistochemical staining for STAT5 of representative malignant tissue with different staining intensities cores. Arrows mark representative nuclear localization. Scale bar = 500 μM.

**Figure 4 life-12-00240-f004:**
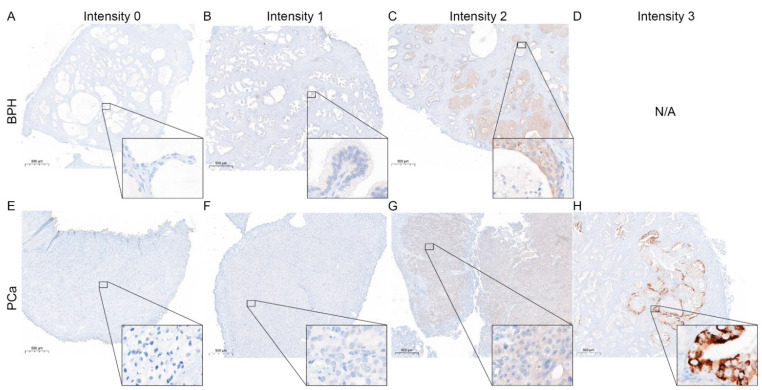
Representative STAT6 staining in BPH and PCa. (**A**–**D**) Immunohistochemical staining for STAT6 of representative benign tissue with different staining intensities cores. Scale bar = 500 μM. (**E**–**H**) Immunohistochemical staining for STAT6 of representative malignant tissue with different staining intensities cores. Scale bar = 500 μM.

**Figure 5 life-12-00240-f005:**
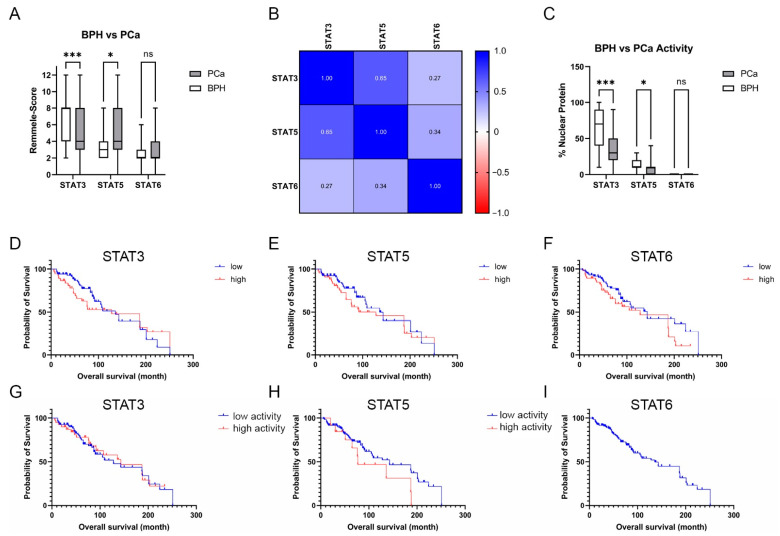
Evaluation of the STAT3, 5, and 6 protein staining in BPH and PCa. (**A**) Evaluation of the expression of STAT3,5, and 6 of BPH and PCa tissue using the Remmele score. Data are shown as box and whisker diagrams (ns: not significant,*: *p* ≤ 0.05, ***: *p* ≤ 0.001). (**B**) Pearson correlation of STAT3, 5, and 6 in PCa tissue. The r-values are displayed in a heat map. (**C**) Evaluation of the % STAT3, 5, and 6 nuclear localization in BPH and PCa tissue as a surrogate for the activity of the transcription factors. Data are shown as box and whisker diagrams (ns—not significant; *: *p* ≤ 0.05, ***: *p* ≤ 0.001). (**D**–**F**) OS analysis of patients with low and high STAT3, 5, or 6 expressions (**D**) Kaplan–Meier curves indicating OS according to the STAT3 expression level of the PCa cohort. The median STAT3-IRS was chosen as the threshold. (**E**) Kaplan–Meier curves indicate OS according to STAT3 expression level. The median STAT5-IRS was selected as the threshold. (**F**) Kaplan–Meier curves indicate OS according to STAT6 expression level. The median STAT6-IRS was chosen as the threshold. (**G**–**I**) OS analysis of patients with low and high STAT3, 5, or 6 activity. (**G**) Kaplan–Meier curves indicating OS according to the STAT3 expression level of the PCa cohort. The median % of nuclear STAT3 was chosen as the threshold. (**H**) Kaplan–Meier curves indicate OS according to STAT3 expression level. The median % of nuclear STAT5 was selected as the threshold. (**I**) Kaplan–Meier curves indicate OS according to STAT6 activity.

**Figure 6 life-12-00240-f006:**
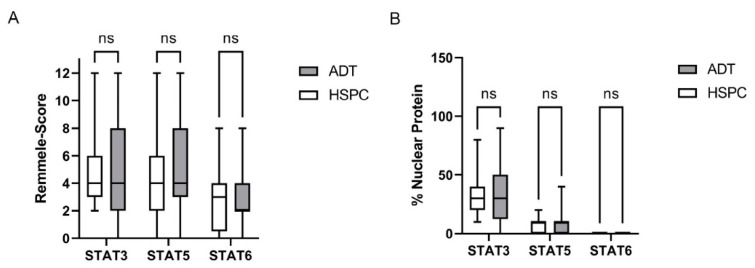
Evaluation of the STAT3, 5, and 6 levels and activity in HSPC and ADT subgroups. (**A**) Evaluation of the expression of STAT3,5, and 6 in PCa tissue of the HSPC and ADT subgroups using the Remmele score. Data are shown as box and whisker diagrams (ns.: not significant). (**B**) Evaluation of the % STAT3, 5, and 6 nuclear localization in PCa tissue of the HSPC and ADT subgroups as a surrogate for the activity of the transcription factors. Data are shown as box and whisker diagrams (ns.: not significant).

**Figure 7 life-12-00240-f007:**
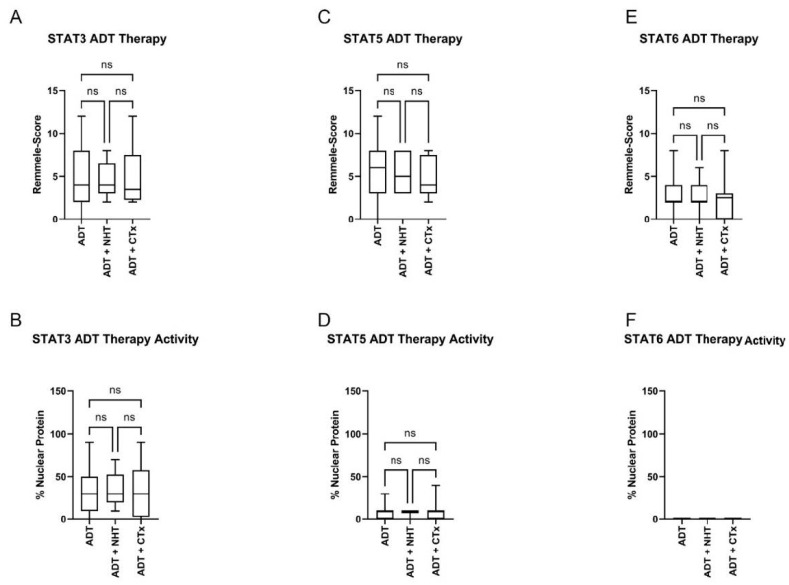
Evaluation of the STAT3, 5, and 6 levels (**A**+**C**+**E**) and activity in HSPC and ADT subgroups (**B**+**D**+**F**). (**A**+**C**+**E**) Evaluation of the expression of STAT3 (**A**), 5 (**C**), and 6 (**E**) in PCa tissue of the HSPC and ADT subgroups using the Remmele score. Data are shown as box and whisker diagrams (ns.: not significant). (**B**+**D**+**F**) Evaluation of the % STAT3 (**B**), 5 (**D**), and 6 (**F**) nuclear localization in PCa tissue of the HSPC and ADT subgroups as a surrogate for the activity of the transcription factors. Data are shown as box and whisker diagrams (ns.: not significant).

**Figure 8 life-12-00240-f008:**
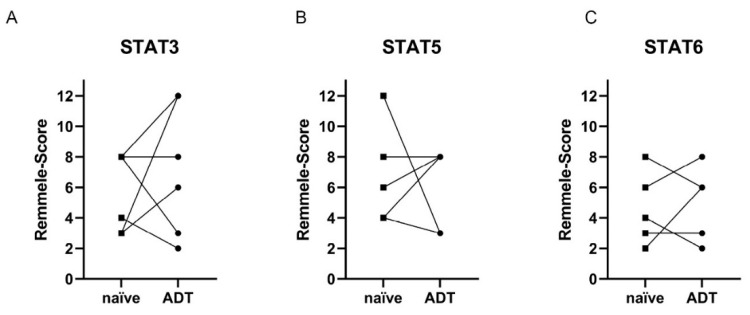
Evaluation of the STAT3, 5, and 6 levels in HSPC (naïve) and ADT subgroups. Evaluation of the expression of STAT3 (**A**), 5 (**B**), and 6 (**C**) in PCa tissue of the HSPC and ADT subgroups using the Remmele score. Individual data points are shown. Corresponding patient data before and under ADT have been connected.

**Figure 9 life-12-00240-f009:**
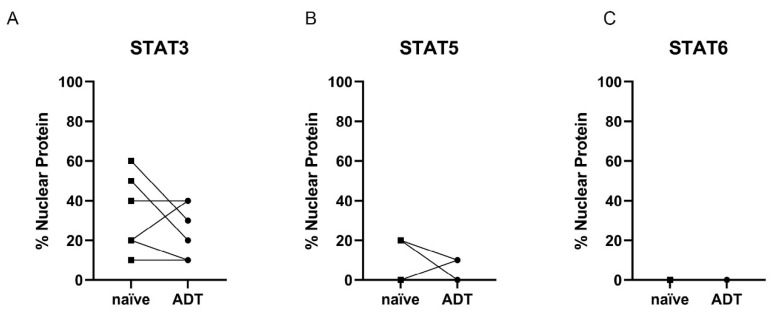
Evaluation of the STAT3, 5, and 6 activity in HSPC (naïve) and ADT subgroups. (**A**+**B**+**C**) Evaluation of the % STAT3 (**A**), 5 (**B**), and 6 (**C**) nuclear localization in PCa tissue of the HSPC and ADT subgroups as a surrogate for the activity of the transcription factors. Individual data points are shown. Corresponding patient data before and under ADT have been connected.

**Table 1 life-12-00240-t001:** Antibodies used in the study.

Name	Company	Lot	Dilution
STAT3 (124H6) Mouse mAb #9139	Cell Signaling Technology, Danvers, MA, USA	16	1:300
Anti-STAT5a + STAT5b antibody [EPR16671-40]	Abcam, Cambridge, United Kingdom	GR3247129-1	1:4000
STAT6 (EP325)	Bio SB, Inc., Santa Barbara, CA, USA	3425QLD05	1:25

**Table 2 life-12-00240-t002:** Baseline characteristics of PCa-patients cohort.

	All	HSPC	mCRPC		
			ADT Only	ADT + NHT	ADT + CTx
Patient Number	97	26	25	33	13
Median age at primary diagnosis, years	73	72	74	71	66
Median PSA at primary diagnosis, ng/mL (Interquartile range IQR)	18 (6.9; 79.0)	6.3 (2.7; 10.2)	15.6 (7.2; 87.6)	49.0 (11.2;128.0)	60.4 (28.7; 92.7)
Neuroendocrine differentiation at primary diagnosis, %	1.0	0.0	0.0	3.0	0.0
Presence of bone metastases at primary diagnosis, %	25.0	3.8	20.0	30	62
Presence of lymph node metastases at primary diagnosis, %	14	12	8.0	18	15
Presence of organ metastases at primary diagnosis, %	1.00	0.00	0.0	3.0	0.0
Median overall survival since the start of primary therapy, months	59	47	81	67	86

## Data Availability

Not applicable.

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
