# Peer review of "Influence of Systemic Therapy on the Expression and Activity of Selected STAT Proteins in Prostate Cancer Tissue"

_life, 2022, doi:10.3390/life12020240_

Round 1
Reviewer 1 Report
In their manuscript, the authors examine the expression and activity of STAT3, STAT5, and STAT6 in PCa patients undergoing systemic therapies including ADT, enzalutamide, abiraterone, and docetaxel.
Of note, most finding of STAT proteins in therapy resistance was demonstrated in vitro, thus I find it as a novelty.
In general, I find the authors have interpreted and presented the relevant results correctly.
Even if the general conclusion says there is no significant increase of expression and activity these 3 proteins after systemic therapy, authors’ work in a convincing way prompts the urge to investigations of these and other STAT proteins in a longitudinal patients cohort to identify factors responsible for the divers influence of system therapy on STAT expression. I think that this is the strength of their story.
In fact, authors’ challenge was to deal with the genetic and cellular heterogeneity of samples. It’s obvious, that a statistical significance of this cohort will be hardly to obtain. One of the options would be enlarging the cohort plus making the filtration of samples including: (a) tumor cellularity, (b) pathologic state and more importantly molecular taxonomy based on the seven gnomically distinct subtypes.
Nevertheless, their work, in my opinion, is still worth to be published.
Minor: In Figure 3, arrows indicating nuclear stains would be helpful.
Author Response
On behalf of all authors, we would like to take this opportunity to express our sincere gratitude to the reviewers who identified areas of our manuscript that needed correction or modification. Their insightful comments have led to an improvement in our manuscript. Below you find the detailed response to the reviewers' comments:
Minor: In Figure 3, arrows indicating nuclear stains would be helpful.
We added arrows for representative examples of nuclear localization into the IHC figures.
Reviewer 2 Report
This study aimed to validate preclinical data concerning the link between STAT3, 5, and 6 and resistance to androgen receptor inhibition and metastasis, using IHC on tissue obtained from patients with prostate cancer before and while under androgen deprivation therapy. The study is interesting, up to date, in the scope of the Journal and might be considered for publication after careful minor revisions.
- Introduction: Please add a sentence on the survival rates of patients with PCa depending on tumor stage
- Methodology: Please use SI units for volume (mL not ml) throughout the manuscript
- Methodology: Please use the same number of decimals for all presented data, and use “.” instead of “,” to present values with decimals
- The last part of the Discussions section needs to be reinforced with a short paragraph on the authors’ point of view on what might be proposed as better models of research in question. This study did not provide conclusive data on correlation with current literature data, thus new models need to be evaluated and proposed.
- Please include a short discusion on study limitations
- Conclusions section:
Please shorten and merge the first sentences:
“STAT proteins are involved in numerous oncogenic processes in PCa. Especially STAT3 and STAT5 have been identified as key players in tumor relapse, metastasis, and therapy resistance. Most of the findings were obtained in preclinical in vitro and in vivo models. Therefore, this study attempted to transfer the preclinical observations to patient material.”
into just 1 sentence, limiting the text only to conclusions directly obtained from the study, and avoiding literature review.
Author Response
On behalf of all authors, we would like to take this opportunity to express our sincere gratitude to the reviewers who identified areas of our manuscript that needed correction or modification. Their insightful comments have led to an improvement in our manuscript. Below you find the detailed response to the reviewers' comments:
- Introduction: Please add a sentence on the survival rates of patients with PCa depending on tumor stage
We added the information in the introduction.
- Methodology: Please use SI units for volume (mL not ml) throughout the manuscript
We replaced "ml" with the SI unit.
- Methodology: Please use the same number of decimals for all presented data, and use "." instead of "," to present values with decimals
Where possible, we unified the number of decimals.
- The last part of the Discussions section needs to be reinforced with a short paragraph on the authors' point of view on what might be proposed as better models of research in question. This study did not provide conclusive data on correlation with current literature data, thus new models need to be evaluated and proposed.
A paragraph about the authors' point of view about preclinical models has been added.
- Please include a short discusion on study limitations
A paragraph about limitations had been added.
- Conclusions section:
Please shorten and merge the first sentences:
"STAT proteins are involved in numerous oncogenic processes in PCa. Especially STAT3 and STAT5 have been identified as key players in tumor relapse, metastasis, and therapy resistance. Most of the findings were obtained in preclinical in vitro and in vivo models. Therefore, this study attempted to transfer the preclinical observations to patient material." into just 1 sentence, limiting the text only to conclusions directly obtained from the study, and avoiding literature review.
We shortened the sentence.
Reviewer 3 Report
This interesting study seeks to address the relationship between the tissue expression of several of the STAT family of transcription factors and survival in prostate cancer patients. The main finding is that although STAT expression (particularly STAT 3 and 5) changes in prostate cancer, there does not appear to be a relationship between expression and survival. This is not clearly stated either in the abstract or in the body or conclusions of the manuscript. The authors have surrounded these results with quite a bit of verbiage that only serves to obfuscate and confuse the reader. Their discussion of this discrepancy with respect to cell culture studies leaves much to be desired and is not clear. I would suggest revising this manuscript in a manner so as to emphasize the main point above and then to discuss the discrepancy in a a more straightforward fashion.
In the matter of methods, it is not indicated what is being presented in several of the box/whisker plots and some are hard to read. I would suggest a sentence or two in the methods describing what is plotted, particularly in Figs. 5,6&7. There are so many boxes, dots and lines that it is hard to see the trees for the forest. I also have a question about this statement:
The IRS-evaluation of STAT3 (Figure 5A) revealed a significant downregulation of 168 STAT3 in PCa (mean IRS ± SD: 7.1±2.5) compared to BPH (mean IRS ± SD: 4.9±3.0).
If the staining intensity and cell number go up from 4.9 to 7.1 is this not enhanced expression rather than decreased expression?
Author Response
On behalf of all authors, we would like to take this opportunity to express our sincere gratitude to the reviewers who identified areas of our manuscript that needed correction or modification. Their insightful comments have led to an improvement in our manuscript. Below you find the detailed response to the reviewers' comments:
This interesting study seeks to address the relationship between the tissue expression of several of the STAT family of transcription factors and survival in prostate cancer patients. The main finding is that although STAT expression (particularly STAT 3 and 5) changes in prostate cancer, there does not appear to be a relationship between expression and survival. This is not clearly stated either in the abstract or in the body or conclusions of the manuscript. The authors have surrounded these results with quite a bit of verbiage that only serves to obfuscate and confuse the reader. Their discussion of this discrepancy with respect to cell culture studies leaves much to be desired and is not clear. I would suggest revising this manuscript in a manner so as to emphasize the main point above and then to discuss the discrepancy in a a more straightforward fashion.
We added the requested statement into the abstract and the conclusion. In addition, we shortened the conclusion and added a summary of the findings after each result section. Therefore, it should be more straightforward. In addition, due to reviewer 2, a paragraph about the cell culture problem has been added. This paragraph also explains the discrepancy between cell culture and patient data.
In the matter of methods, it is not indicated what is being presented in several of the box/whisker plots and some are hard to read. I would suggest a sentence or two in the methods describing what is plotted, particularly in Figs. 5,6&7. There are so many boxes, dots and lines that it is hard to see the trees for the forest.
All IRS data with the expectation of the longitudinal patient cohort data has been shown as box and whisker diagrams showing all data points. We removed all data points. Therefore, we hope the data is now more precise.
I also have a question about this statement:
The IRS-evaluation of STAT3 (Figure 5A) revealed a significant downregulation of 168 STAT3 in PCa (mean IRS ± SD: 7.1±2.5) compared to BPH (mean IRS ± SD: 4.9±3.0).
If the staining intensity and cell number go up from 4.9 to 7.1 is this not enhanced expression rather than decreased expression?
We want to apologize for this mistake and thank the reviewer for stating it out. The numbers should be exchanged. We corrected the error in our manuscript.
This manuscript is a resubmission of an earlier submission. The following is a list of the peer review reports and author responses from that submission.
Round 1
Reviewer 1 Report
This study reports expression levels of STAT3, STAT5 and STAT6 in prostate cancer compared to BPH and determines expression changes of these proteins after treatment with androgen pathway inhibitors and their potential association with patient outcomes prediction.
The immunohistochemistry staining results shown in Figures 3 – 5 are of high quality. While there is a need for additional studies on STAT proteins in prostate cancer, the study in this manuscript does not provide a major advance in the filed because of several shortcomings:
- The sample size is small compared to the sources of variability in the patient population. All ADT drug treatments were combined without stratification based on the exact drug used. It would be helpful to only enroll patients treated with the same category of drug. More information about treatment of patients related to the type androgen inhibitory treatment should be provided.
- The measurement of total STAT protein may not represent the active, nuclear STAT protein. This could lead to a bias within the small cohort size.
- In S1, it is unclear how STAT activity was determined. The lack of correlation could be due to measuring total STAT and not nuclear STAT
- The activity of STAT proteins is regulated by phosphorylation. The role of phosphorylation in promoting nuclear localization and transcriptional regulation has been extensively analyzed in cell lines. Phosphoantibodies can be used for staining of FFPE tissues from patients. While there has been a concern of rapid dephosphorylation during tissue collection, functionally important phosphorylation sites are stabilized and presumably prevented from de-phosphorylation through binding of proteins that contain specific domains, such as SH2 domains. After excluding false negative cases due to insufficient tissue quality, staining profiles of pSTAT proteins can be compared to STAT activity based on RNA expression profiles.
- Image analysis should be used to quantify nuclear and cytoplasmic protein expression and to deal with tumor heterogeneity.
Reviewer 2 Report
The authors have set up a very well controlled immunohistochemistry study of STAT3, 5 and 6 in different stages of prostate cancer. The question on clinical relevance of STAT expression in PCa is an older one and early clinical trials with inhibitors have not delivered, which has been ascribed to heterogeneity in the patient population and hence a need for more personalized approaches by identifying possible patient groups that could benefit from targeting these transcription factors. However, it remains to be seen whether this is the case. We see in this study that there is indeed heterogeneity in STAT3 and 5 levels (Fig. 5C) and how they change in response to treatment (Fig. 8,9).
My major comment is the fact that this is a single center study with limited longitudinal data on only STAT3, 5 and 6 levels, which does not allow very strong conclusions. I understand the difficulties of setting up longitudinal studies in PCa. However, due to the very limited numbers, the longitudinal data in figs 8 and 9 can only to be considered anecdotal or pilot data. This leads to a very weak final statement in the abstract (‘…justifies further investigations of STAT proteins as therapeutic targets in translational research…’). One could turn around this argument and state that it would be better to first invest in a deeper study of STAT3 and 5 in clinical samples. Unfortunately, the work in this manuscript remains descriptive and in parts only confirmatory (published work is cited correctly in this manuscript).
Additional data on the same clinical samples describing IHC validation of possible STAT up- or downstream effects like immune responses, mitogenesis, cell survival, cell growth (see first sentence of te discussion), could make this work stronger.